# National Quality Infrastructure System and Its Application Progress in Photovoltaic Industry

Rui Sun [1], Hua-Feng Xiao [1,*], Chen-Hui Niu [2], Qing-Wei Cao [2] and Zhong-Yuan Yao [2]

1  School of Electrical Engineering, Southeast University, Nanjing 210096, China; sunrui_sr@seu.edu.cn
2  Huaneng International Power Jiangsu Energy Development Co., Ltd., Nanjing 210015, China; ch_niu@jsgs.chng.com.cn (C.-H.N.); qw_cao@jsgs.chng.com.cn (Q.-W.C.); yaozhongyuan@jsgs.chng.com.cn (Z.-Y.Y.)
*  Correspondence: xiaohf@seu.edu.cn

**Abstract:** With the rapid development of economic construction, National Quality Infrastructure (NQI) has received increasing attention from countries and international organizations. NQI is a comprehensive system and capacity building, which plays a key role in promoting healthy and sustainable economic and social development. However, the photovoltaic industry has not yet established an NQI system and lacks an overall quality supervision mechanism. This will hinder the comprehensive development of the photovoltaic industry in terms of standards, metrology, conformity assessment, etc. In this paper, first, the concept and overall framework NQI is sorted out; the three most important elements of NQI are pointed out. Then, on the basis of fully explaining the NQI, an NQI system for the photovoltaic industry is established for the first time, and the construction of the NQI elements of the photovoltaic industry internationally is sorted out in detail. Finally, the possible solutions to the problems existing in the overall construction of the NQI system are proposed. Points for improvement are listed for each element of the NQI system for the photovoltaic industry.

**Keywords:** National Quality Infrastructure; conformity assessment; standards; certification and accreditation; inspection and testing; photovoltaic industry

## 1. Introduction

The level of product quality is related to the safety of people's lives, social stability and even the country's international influence. Quality involves the entire process from production to government supervision, and has always been an important issue of concern to countries all over the world. "National Quality Infrastructure (NQI)"is hailed as an effective measure to solve quality problems. In 2007, Physikalisch Technischen Bundesanstalt (PTB) and other institutions jointly published "The Ultimate Answer to Global Quality Problems—National Quality Infrastructure" [1]. It is mentioned in the book that in order to obtain good quality, it is necessary to achieve coordination in several interrelated aspects such as measurement, standards, testing, accreditation, and certification.

The concept of NQI was first proposed by PTB in 2002, and was officially released to the world by the International Trade Centre (ITC) in 2005, and has been widely recognized by the international community. The World Bank released two special reports on quality in 2011 and 2013, focusing on the important role of NQI in promoting international trade, technological development, and safeguarding consumer rights. At the same time, the United Nations Industrial Development Organization (UNIDO) and other international and national organizations have promoted the NQI concepts and projects to countries in Africa, America, and Asia. The construction of China's NQI started in 2012. In 2014, the Science and Technology Department of AQSIQ (State General Administration of the People's Republic of China for Quality Supervision and Inspection and Quarantine) put forward the NQI special plan for the first time. In 2016, NQI was included in China's "National Science and Technology Innovation "13th Five-Year Plan" [2]. In 2017, CPC

Central Committee and State Council issued "Guiding Opinions on Carrying out Quality Improvement Actions" [3] and other documents, which gave guiding opinions on quality improvement work, marking that China's economic development has entered an era of quality. At the same time, various local governments in China have also introduced corresponding policies and measures to accelerate the construction of NQI, and relevant companies have also actively established NQI "one-stop" public service platforms. At present, China has initially formed an NQI system, including legal and regulatory system, management system, and technical system.

Government management departments, research institutions and scholars all over the world have carried out in-depth research on the NQI system. Liu and Hu [4] divide the development of NQI into three stages, and trace the origin of the rise of NQI. The needs and thinking about NQI have sprouted in the middle of the 18th century, and they are constantly exploring and developing in social changes. Jiang [5] summarizes the practical application of NQI in the world. Quantitative evaluation of the development of NQI has also become a hot topic in various countries. The main methods include questionnaire survey method, comparative experiment method, applied statistics method, modeling simulation method, etc. PTB and UNIDO first proposed the framework, influencing factors and evaluation methods of NQI [6]. Among them, PTB uses the World Trade Organization (WTO) member countries as samples and uses composite indicators for assessment of the current state of NQI development [7]. Moljevic [8] establishes feedback among the 19 elements related to various aspects of NQI, such as the capital chain, the level of management and control, the ability to improve quality, the ability to achieve quality, and the level of production technology. By adjusting the input of parameters and various factors, a dynamic model of the regional quality system to study the influence of quality on regional development is constructed. It is concluded that the quality level is mainly affected by the distribution of the testing laboratory network and the development of GDP per capita. Choi et al. [9] established an NQI evaluation system with standards, conformity assessment, and metrology for 4: 3: 3 by drawing on the theory of total quality management. The United States [10], Sweden [11], the Philippines [12,13] and other countries have carried out the implementation of NQI projects and issued NQI reports based on their own national conditions. Literature [14] discusses the importance of NQI. Xu [15] analyzes the development status of China's NQI technology system, as well as the opportunities and challenges it faces. The integrated development of NQI is the current development direction of quality construction, and all elements of NQI are required to be interconnected, complement each other, and develop in a coordinated manner. Chen and Deng [16] sorted out the factors restricting the integrated development of NQI, and put forward policy recommendations to improve the integrated development of NQI.

Based on the above documents, NQI has now been widely recognized globally, and many countries have also established an NQI system that takes the overall situation in accordance with their own conditions. However, the above literature focuses on the origin, development and application of NQI, and lacks detailed introduction to the system and components of NQI. This gap in the literature causes readers to experience a lack of systematic understanding of the NQI system and its significance.

With the decline in the cost of photovoltaic equipment in recent years, the installed capacity of photovoltaics has increased rapidly, and the relative perfection of the photovoltaic industry chain also ensures that photovoltaic energy is expected to become the main energy source in the future. According to data from the International Energy Agency, as of the end of 2020, the world's cumulative installed photovoltaic capacity was 760.4 GW. There were more than 1 GW of new installed capacity in 20 countries, of which China, the European Union and the United States have newly installed capacity of 48.2 GW, 19.6 GW, and 19.6 GW, respectively, ranking the top three in the world. According to Solar Power, Europe's forecast in the "Global Market Outlook for Solar Power" released on 1 September 2021, the global cumulative new photovoltaic capacity will reach 1000 GW to 1097 GW from 2021 to 2025. At present, China has become the world's leader in solar power generation,

with the world's largest photovoltaic power generation industry chain, the largest application market, the largest investment country, the most invention and application patents, and the largest exporter. The explosive growth of the photovoltaic market and the proposal of carbon emission targets in various countries mark that the photovoltaic industry has entered a stage of high-quality development. However, there are still some problems in the development of the photovoltaic industry, especially distributed photovoltaic, which has market investment chaos and frequent accidents, such as photovoltaic array accidents, electrical accidents, and safety accidents. These problems can be solved by market supervision, design specifications, standard research, certification and testing, etc., and coordinated through the establishment of an NQI system for the photovoltaic industry. However, the NQI concept has not been popularized in the photovoltaic industry. The photovoltaic industry has not established an NQI system, has not implemented a quality management mechanism for the entire industry chain, and has not yet established a national NQI unified platform. All this makes the photovoltaic industry fail to achieve overall coordination at the national, industry and market levels, and develops in a fragmented situation in terms of standards, conformity assessment, metrology, and market supervision. It is urgent for government departments of various countries to actively establish the NQI management system of the photovoltaic industry to improve the photovoltaic standard system, testing and certification system, and ensure the market supervision mechanism, etc.

The rest of the paper is structured as follows: Section 2 introduces the composition of NQI and the meaning of each component in detail. The meaning of each component of NQI is explained in detail, so that readers have a comprehensive and detailed understanding of the NQI system. Then, the NQI system of the photovoltaic industry is first established and the development of the various elements of the NQI in the photovoltaic industry is analyzed in Section 3. The establishment of the NQI system of the photovoltaic industry will help to comprehensively and systematically inspect the quality construction of the photovoltaic industry. In Section 4, for countries currently building NQI systems, such as China, possible solutions to some problems in the process of NQI construction are proposed. Possible development suggestions are also put forward for the construction of the NQI system in the photovoltaic industry. This will try to help various countries to establish and improve the NQI system and improve the quality construction of the photovoltaic industry.

## 2. National Quality Infrastructure

NQI is an internationally accepted concept [17], which mainly consists of standards, metrology, and conformity assessment (including certification and accreditation, inspection and testing). It meets the quality needs of users and the technical needs of regulators and producers. The technology chain composed of these three pillars is an important technical means for the government and enterprises to increase productivity, maintain safety, improve quality, protect consumer rights, protect the environment, and maintain people's lives and health [18]. It can effectively guarantee social security, international trade and sustainable development.

### 2.1. NQI Core Elements

With the coordinated development of the world economy, NQI's globalization needs have become increasingly strong and continue to be given more meaning. In 2017, the DCMAS network and the World Bank jointly gave the latest definition of a quality infrastructure system [4]. NQI is described as "a system composed of public and non-governmental organizations and policies, relevant laws, regulations, and practices required to support and improve the quality, safety, and environmental protection of products, services, and processes. The NQI system relies on metrology, standards, accreditation (listed out from the conformity assessment), conformity assessment and market surveillance. "This new definition adds market surveillance requirements to the original definition. It can be seen from the definition that NQI has five elements, namely, standard, metrology, conformity assessment, accreditation (taken out from conformity assessment), and market supervision,

as shown in Figure 1. The conformity assessment also contains four elements of certification, accreditation, inspection and testing.

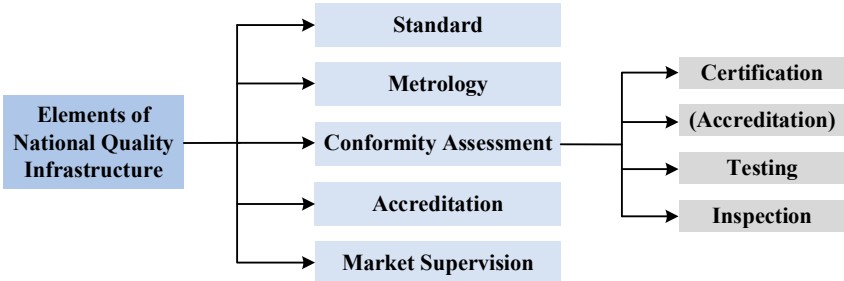

**Figure 1.** Elements of National Quality Infrastructure.

Among the five elements of NQI, the core elements are standards, measurement and conformity assessment. Metrology is the basis for quality control, standards lead quality improvement, conformity assessment controls quality and builds quality trust. The three forms a complete technical chain, as shown in Figure 2. They interact and promote each other, and jointly support the development of quality. Specifically, metrology solves the problem of accurate measurement. The value of quality shall be uniformly regulated by the standard. How well the standard is implemented needs to be judged by conformity assessment.

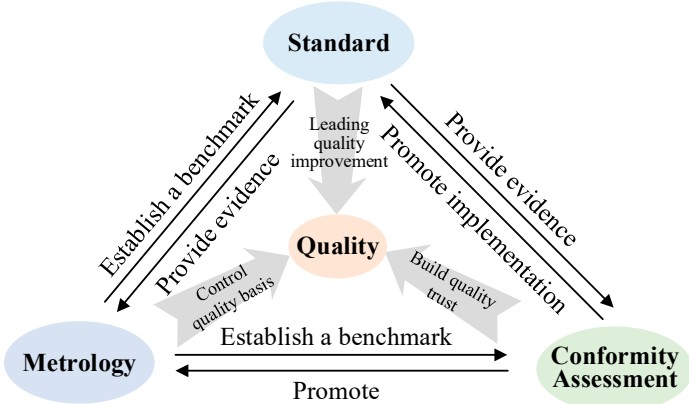

**Figure 2.** The relationship between the three elements of NQI.

### 2.2. Metrology

Metrology is a variety of measurement activities carried out to achieve unity and accurate measurement, which is the prerequisite and foundation of NQI. Metrology generally has the following four characteristics: accuracy, consistency, traceability, and legality [19]. Accuracy means that the measurement result is accurate within a certain allowable error range. Consistency means that no matter how different the measurement time, location, measurement method, or measuring instrument are, as long as they are in the same measurement unit, the measurement results should be consistent within the allowable error range. Traceability means that the measurement results can be attributed to the original measurement instrument or measurement benchmark through traceability comparison, which ensures the consistency and accuracy of metrology. Legality refers to the requirements of national laws on measurement accuracy, which can effectively guarantee social fairness and justice.

The metric system appeared during the French Revolution, which made European politicians realize that a unified measurement system can strengthen national unity. In modern times, Europe has established a unified measurement system—the International

System of Units SI, which has been widely used. China has a long history of measurement, and it has been called "weights and measures" in the primitive society. After the founding of New China, China actively updated, enriched, and developed the measurement system. In 1985, the National People's Congress of China voted and passed the "Metrology Law of the People's Republic of China" [20]. The bill has been revised and supplemented many times. At present, China has formed a legal and regulatory system with metrological law as the basic law, and a variety of technical laws and government regulations supplemented. A clear and complete measurement system has been formed, which provides an effective guarantee for China's metrology work.

With the development of a science- and technology-based society, new theories have promoted the development of metrology science, especially in the field of quantum metrology. Quantum metrology has attracted wide attention from researchers from various countries due to its good security, repeatability, high accuracy, and avoidance of transmission errors. In 2013, China cooperated with foreign research groups to develop quantum measurement detection devices, which greatly promoted the development of China's metrology and related high-tech industries.

### 2.3. Standard

In essence, standards are unified regulations for repetitive things and concepts. Their task is to standardize and adjust various market economy-related objects, and is the guarantee and basis for quality. As a normative document, the standard has the following characteristics: (1) allows repeated use and common use; (2) is formulated by an authoritative organization in accordance with the process, and is issued after consultation; (3) specifies the scope of application [21]. The process and activities of formulating, publishing and implementing standards are called standardization. Standardization is the primary means to promote technological innovation, an important way to reduce transaction costs, a "passport" to the international market [22], and an important technical support for national economic and social development.

Due to the continuous development of the international market, the demand for international standards continues to increase. International standards have become tools to support global trade and manufacturing. The International Electrotechnical Commission (IEC) was established in the United Kingdom in 1906 [23]. In 1947, the International Organization for Standardization (ISO) was established. Member states have also established authoritative standards organizations, representing their own countries to participate in the conference and designate standards suitable for their own national conditions. Since then, international standardization and national standardization have flourished. China is also actively participating in the construction of the international standard system and national standards. After the founding of New China, China State Bureau of Standards (CSBS) was established to carry out standardization work, and began to compile national standards and industry standards. Later, the ISO quality management system was introduced, and The State Bureau of Quality and Technical Supervision was established jointly with State Administration of metrology of China, and the relevant laws and regulations for the implementation of standardization were promulgated. At present, China has formed a standard system with national standards as the mainstay, with industry standards, local standards, and corporate standards developing together and complementing each other. China has also participated in the formulation of 297 international standards. At present, the proportion of international standards developed and promulgated by China has reached 1.9% of the total number of standards promulgated by ISO and IEC.

### 2.4. Conformity Assessment

Conformity assessment is a means of building trust in the market. It judges whether related products and services meet expectations in accordance with relevant standards, regulations and norms. ISO/IEC 17000 "Conformity Assessment—Vocabulary and General Principles "defines Conformity Assessment as "the verification that the requirements re-

lated to products, processes, systems, personnel or institutions are met". That is, consumers or users have certain expectations for the quality, economy, environmental protection, reliability, safety, compatibility, operability, efficiency, and effectiveness of products and services. Conformity assessment is used to prove that these features meet the requirements of relevant standards, regulations or other specifications.

Conformity assessment is divided into four methods: testing, inspection, certification, and accreditation. ISO/IEC 17000 "Conformity Assessment—Vocabulary and General Principles" defines the four methods, respectively. Testing refers to the determination of one or more characteristics of an object of conformity assessment, according to a procedure. That is, a series of tests are performed on the evaluation objects in accordance with relevant standards or specifications, and the evaluation results are the test data. Inspection refers to examination of an object of conformity assessment and determination of its conformity with detailed requirements or, on the basis of professional judgement, with general requirements. To put it simply, the inspection not only provides the results, but also compares it with the specified requirements to make a judgement whether it is qualified or not. Certification refers to third-party attestation related to an object of conformity assessment, with the exception of accreditation. That is to say, the procedure for a third party to issue a written certification that its products, processes or services meet the specified requirements, which is essentially a form of credit guarantee. Among them, third-party institutions refer to institutions that are independent of the supply and demand sides. Accreditation refers to third-party attestation related to a conformity assessment body, conveying formal demonstration of its competence, impartiality, and consistent operation in performing specific conformity assessment activities. The acquisition of accreditation needs to pass a competency assessment and receive appropriate supervision. Recognition, certification, and issuance of socially useful inspection and test reports can all be called third-party conformity assessments.

The first country to implement a certification system in the world was the United Kingdom. In 1903, the UK registered the BS mark for certification. In the 1950s, the certification system gained popularity in industrially developed countries. In 1970, ISO established the "Certification Committee (CERTICO)", and in 1985 it was renamed the International Organization for Standardization/Committee on Conformity Assessment (CASCO). Its main responsibility is to coordinate the certification systems of various countries from a technical perspective and establish an international quality certification system. China's conformity assessment work started late. In 2001, China established the Certification and Accreditation Administration of the People's Republic of China (CNCA) for unified management, supervision and coordination of certification and accreditation across the country. In 2002, the China National Accreditation Service for Conformity Assessment (CNAS) was established. CNAS has approved and authorized the establishment of multiple national accreditation agencies, which are responsible for the accreditation of certification agencies, laboratories and inspection agencies. As of August 2021, the number of accredited inspection institutions in China has exceeded 700, and the overall number is at the leading level in the Asia-Pacific region. China's certification industry has also achieved rapid development under the regulations of various certification agencies. As of August 2021, China has issued 2.82 million valid certification certificates, more than 300 million test and inspection reports, and the number of certified companies has reached 830,000. In 2004, CNAS joined the Asia-Pacific Laboratory Accreditation Cooperation (APLAC) Agreement on Mutual Recognition of Inspection Agencies, achieving international mutual recognition, gaining a good reputation in the international community, and building a trust platform for Chinese inspection agencies.

*2.5. Integrated NQI Development*

The integrated development of NQI refers to the establishment and implementation of an institutional framework for the various elements of the NQI, and the overall coordination through government departments and related agencies. The integrated development of

NQI can eliminate barriers between departments and elements, reduce the approval process, promote joint collaboration, realize unified management and information sharing, and ensure the high-quality development of NQI [24].

At present, China has actively developed NQI multi-element integrated services. In 2020, China State Administration for Market Regulation issued the "Opinions on Vigorously Developing "One-Stop"Services for Quality Infrastructure", proposing to develop NQI "one-stop"services in accordance with economic laws. In December of the same year, Yantai City of China took the lead in building the NQI+ service cloud platform, creating a Yantai model of "online and offline integration"services, and boosting the high-quality development of enterprises through the integration, sharing and development of quality infrastructure [25].

## 3. Applications of NQI in Photovoltaic Industry

### 3.1. Establishment of NQI System for Photovoltaic Industry

The NQI system of the photovoltaic industry can be divided into metrology, standard, conformity assessment and market supervision according to the elements of NQI. The form of the NQI system for the photovoltaic industry is shown in Figure 3. The metrology in the photovoltaic industry includes basic metrology and energy metrology. For the standards of the photovoltaic industry, there is an IEC photovoltaic standard system internationally, and each country also has its own standard system. For example, China has a comprehensive standardized technology system for the photovoltaic industry. Photovoltaic conformity assessment mainly refers to the certification and testing work carried out by institutions that have obtained national accreditation. The market supervision of photovoltaics is mainly aimed at the cost and market supervision of project construction. The following will elaborate on the application progress of the four elements of NQI in the photovoltaic industry.

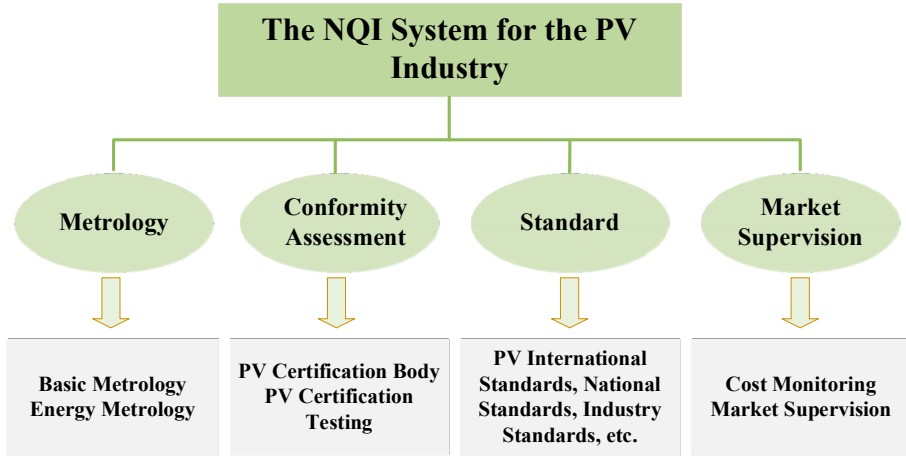

**Figure 3.** Establish an NQI system for photovoltaic industry.

The construction of an NQI system for the photovoltaic industry can help the photovoltaic industry have an overall management and restraint mechanism, and can also help the photovoltaic industry develop in an all-round way. However, attention should also be paid to the rational construction of the system to avoid complex and redundant systems.

### 3.2. Metrology in Photovoltaic Industry

The yield and conversion efficiency of the photovoltaic industry are very important for battery manufacturers. There is also an increasing demand for measurement and testing of technical indicators of photovoltaic products. The entire industrial chain of the photovoltaic industry, including crystalline silicon ingots, silicon wafers, photovoltaic cells, and system

installation, requires the participation of metrology. Metrology in the photovoltaic industry can be divided into basic metrology and energy metrology.

The basic measurement capability determines the energy conversion capability of photovoltaic products. The measurement of solar cells is mainly to measure the size, photoelectric characteristic parameters, and battery photoelectric conversion efficiency. The measuring instruments used include solar cell I-V characteristic measurement system, solar cell sorting machine, solar simulator, cell EL tester, etc. Mainly, these instruments carry out a series of measurements such as short-circuit current, open circuit voltage, short-circuit current density, light intensity, battery temperature, absolute reflectance, and relative transmittance [26]. High-efficiency battery modules can ensure stable and long-lasting power generation. The measurement and inspection of photovoltaic modules mainly include conversion rate, light transmittance, reflectance, etc. In addition, it is necessary to calibrate and trace the measurement equipment of photovoltaic products. Through accurate metrology of the photovoltaic industry chain, testing institutions can guarantee the overall quality of the photovoltaic industry chain.

Among PTB's research institutions, the four competence centers are particularly important, one of which is the photovoltaic metrology center. PTB's photovoltaic center can obtain the world's best measurement uncertainty, providing a safe and stable measurement basis for photoelectric metrology. Many German companies in advanced positions in the photovoltaic field have obtained PTB calibration services and provide strong guarantees for photovoltaic measurement technology. PTB's calibration capability in the field of photovoltaic cells has always maintained an internationally leading position. It can achieve full calibration in the area of solar modules and all influencing variables related to production capacity [27], and can calibrate standard solar cells under user-specified test conditions, and is traceable to the international Standard radiance [28]. China's local metrology departments and domestic testing centers have also increased their attention toward photovoltaic metrology systems. In 2019, the China Metrology Institute has established a photovoltaic value traceability system that can be directly traced to the internationally recognized photovoltaic power standard for cryogenic radiometers with the highest accuracy, which has improved the level of uncertainty in the calibration of photovoltaic modules, and its measurement service has been recognized by CNAS. The China Photovoltaic Industry Measurement and Testing Center went abroad for the first time in 2015 to carry out calibration business.

As the grid-connected, photovoltaic capacity increases, the electric energy metering of photovoltaic power generation systems is also particularly important. The harmonics generated by photovoltaic grid-connected systems will cause measurement errors in current transformers and metering errors. Especially for the large-scale construction of distributed photovoltaics such as rooftop photovoltaics, power electronic components such as inverters will be connected on a large scale. In addition, the poor quality of some inverters has caused the harmonic pollution of the grid and seriously exceeded the standard [29], which caused the ratio and angle differences of the current transformer to change, and the transfer characteristics became worse, making the power meter and electric energy metering smaller [30]. This will affect the economic benefits and market fairness of power users. To solve these problems, scholars from various countries and institutions have carried out related research at present. Naik et al. [31] and Wu et al. [32] have simplified inverters, photovoltaic modules, filters and other devices to obtain equivalent mathematical models, and analyzed their harmonic disturbances. Teng et al. [33] have proposed a harmonic power measurement algorithm based on wavelet packet decomposition and reconstruction. Dou et al. [34] have analyzed the characteristics of harmonics and voltage fluctuations caused by distributed photovoltaic grid-connected, and established a measurement correction module for the characteristics of harmonics and voltage fluctuations.

### 3.3. Conformity Assessment for Photovoltaic Industry

The entire industrial chain and life cycle of photovoltaics (including the decommissioning and recycling of photovoltaic equipment) are subject to conformity assessment. A complete photovoltaic certification system includes three aspects: (1) certification standards, including safety regulations and grid-connected requirements; (2) nationally recognized certification bodies [35]; (3) testing laboratories.

Photovoltaic certification is an assessment based on a series of test results. Photovoltaic testing is a laboratory and outdoor testing performed by the photovoltaic industry in accordance with prescribed methods and procedures to verify whether the final performance of products, raw materials, processes, and power stations meet industry standards. Classified according to the scope of the photovoltaic supply chain, there are raw material testing, online process testing, photovoltaic module testing, system testing, power station and grid connection testing, process equipment testing, etc. Raw material testing mainly refers to the testing of the performance of raw materials for the components of the photovoltaic industry and the components of the system. On-line process testing refers to testing performed in order to monitor product quality during the production process, such as on-line sorting of cells. Photovoltaic module testing means that the core components meet performance and safety standards. System testing includes performance and safety testing of various components such as inverters and combiner boxes. Power station and grid-connected testing means that photovoltaic modules and systems are installed outdoors to be debugged and performance tested, and their power-generation performance and power-generation quality must be tested before and after becoming grid-connected. Process equipment testing refers to the testing of production processes and production equipment.

The explosive growth of distributed photovoltaics and the rapid increase in installed capacity both pose challenges to the business capabilities of the inspection industry, and make photovoltaic grid-connected inspection the focus of research in various countries. Grid-connected detection includes active and reactive power detection, power quality detection, low voltage ride through detection, grid-connected inverter detection, anti-islanding protection detection, etc. Related detection technologies include field detection technology [36], grid-connected system current detection technology [37], and maximum power point tracking (MPPT) technology [38]. Among them, the grid-connected inverter connects the photovoltaic system and the grid, and is a key equipment to ensure the long-term reliable operation of the power station and improve the return on investment of the project. The inverter is also responsible for the intelligent control of the entire photovoltaic system, and functions such as monitoring and adjusting the state of the system. It is an important part of grid-connection inspection. IEC has issued a series of photovoltaic grid-connected standards represented by IEC 61727-2004 "Photovoltaic (PV) systems—Characteristics of the utility interface", which elaborated on the content and test procedures of photovoltaic grid-connected testing. Based on the IEEE 929-2000 "IEEE Recommended Practice for Utility Interface of Photovoltaic (PV) Systems", the United States has formulated standards for connecting optical systems to the grid. Due to the distributed characteristics of photovoltaic power plants in European and American countries, the United States mainly refers to standards such as IEEE 1547-2003 "Standard for Interconnecting Distributed Resources with Electric Power Systems" to perform testing. China has also issued GB/T 37409-2019 "Testing specification for photovoltaic grid-connected inverter", which regulates the testing procedures and equipment. International research on grid-connected inverter detection is currently focused on islanding detection [39,40]. In addition, in large-scale, distributed, photovoltaic grid-connected scenarios, non-isolated grid-connected inverters are often used to control costs and improve efficiency. Therefore, it is also necessary to pay attention to the problems of collected leakage current and collected DC components when a large number of non-isolated grid-connected inverters are connected to the grid. It is necessary to lower the limit on the existing testing standards to ensure safety and reliability of power quality.

### 3.3.1. Conformity Assessment Agencies around the World

Both Europe and the United States require that all photovoltaic modules and inverters must pass inspection and certification. After years of development and accumulation, some developed countries have formed a relatively complete certification system for the photovoltaic market. But for developing countries, such as China, the photovoltaic industry has experienced a period of blind development. With the rapidly increasing installation scale, the product quality of hundreds of manufacturers is uneven. Due to concerns about product quality, banks and insurance companies are unwilling to finance and underwrite power plants, which hinders the large-scale development of the photovoltaic industry [41]. In July 2009, China's Ministry of Finance, Ministry of Science and Technology and National Energy Administration jointly issued the "Notice on the Implementation of the Golden Sun Demonstration Project". Major equipment such as photovoltaic modules, controllers, inverters, and storage batteries used in projects supported by financial subsidy funds must be certified by a national certification agency. After market surveys, it is found that the quality of photovoltaic products certified by Golden Sun has improved significantly. The product is safer, more reliable and durable [42]. Since then, the certification system has been gradually promoted in China. In February 2014, the National Energy Administration of China and the CNCA jointly issued the "Implementation Opinions on Strengthening the Testing and Certification of Photovoltaic Products" [43] to implement mandatory testing and certification nationwide. It is also required that photovoltaic power stations that are connected to the grid and receive subsidies must use photovoltaic products from certification agencies approved by the CNCA. Countries such as India and Bangladesh also have compulsory certification requirements for photovoltaic products entering the market.

Some foreign certification agencies, and certification agencies approved by the CNCA, and their businesses, are shown in Table 1.

**Table 1.** Major PV certification agencies.

| Index | Certification Body | Country | Standard Basis | Main Business |
|---|---|---|---|---|
| 1 | Underwriter Laboratories (UL) | USA | UL 61703\UL 1741\IEC 61215\IEC 61646\IEC 61730\EN 61730, etc. | Cover the entire life cycle of the product, including aspects such as research and development, design, production, use and subsequent maintenance. Cover almost all components of the PV system and formulate standards for PV products. |
| 2 | Japan Electrical Safety and Environment Technology Laboratories | Japan | IEC 61730\JISC 61215\JISC 8993 | Certification of crystalline silicon and thin-film PV modules, and certification of operation and maintenance of PV power generation systems. |
| 3 | Intertek | UK | EN 50548\EN 50521\P10-0003\UL 3730\UL 3703\UL 2703, etc. | PV module product compliance testing and certification, PV product performance testing, PV component system balance, solar thermal product testing, solar cell bank subsidies. |
| 4 | VDE Testing and Certification Institute | Germany | IEC 62103\IEC 61646\EN 61000-6-2/VDE 0100\VDE 0126 | Product testing covers complete PV systems, PV modules, power inverters, installation systems, connectors and cables. Services include the following: safety testing, environmental testing, on-site compliance monitoring/inspection according to VDE and IEC standards. |

**Table 1.** *Cont.*

| Index | Certification Body | Country | Standard Basis | Main Business |
|---|---|---|---|---|
| 5 | TÜV SÜD | Germany | ULC/ORD-C1703\UL 1703\IEC 61730\IEC 61215\IEC 61646\EN 1090-2\EN 1090-3 | Provide a complete set of services from solar power plant design review to follow-up routine maintenance inspection, including PV system balance module testing and certification, PV module testing and certification, solar performance evaluation, PV power plant certification, construction project monitoring, etc. |
| 6 | TÜV Rheinland | Germany | UL 9540\UL 1741\CEC Guidelines\IEC 61683\EN 50530 et.al | Provide testing and certification of PV modules and components, supply chain management, and independent PV power station design services. Testable products include ground-use crystalline silicon battery modules, thin-film solar battery modules, concentrating solar battery modules, controllers, inverters, off-grid systems, grid-connected systems, etc. |
| 7 | China Quality Certification Centre (CQC) | China | IEC 62124\IEC 62125\GB 29196\GB 19964\NB 32004\CNCA 0004, etc. | Committed to building a national-level public service platform integrating solar energy (PV, solar thermal) testing, certification, and scientific research. The main business includes the certification of solar PV products (PV modules, PV inverters, PV combiner boxes, energy storage batteries, etc.), and supervision of the manufacturing of key PV components (PV modules, PV inverters, box transformers and main transformers, etc.). |
| 8 | China Classification Society Certification (CCSC) | China | IEC 61215\IEC 61730\GB 29848\GB 31034\NB 32004\CNCA 0003\CNCA 0002, etc. | The service covers the entire industrial chain from raw materials, key equipment of PV power plants to PV power plants. Provide third-party technical services such as PV station address and optical resource evaluation, design evaluation and optimization, power station quality and performance testing (power station acceptance, follow-up evaluation, follow-up monitoring), power station operation and maintenance capability evaluation and certification. |
| 9 | Ceprei Certification (Guangzhou) | China | IEC 62446\NB 32004, etc. | Certification of grid-connected inverters and grid-connected PV systems |
| 10 | Noah Testing Certification Group (NOA) | China | IEC 62124\IEC 62446\IEC 17025\GB 2828, etc. | Provide on-site inspection of PV modules for large-scale ground PV power stations, distributed PV power stations, and household PV power stations, provide full life cycle quality control services such as PV grid-connected inverters, combiner boxes, batteries, independent PV system testing and completion acceptance, etc., and issue third-party testing data and analysis reports. |

**Table 1.** *Cont.*

| Index | Certification Body | Country | Standard Basis | Main Business |
|-------|-------------------|---------|----------------|---------------|
| 11 | China Testing and Certification International Group (CTC) | China | IEC 61646\IEC 61730\IEC 61724\GB 18911\GB 20047, etc. | The testing capabilities cover the entire PV industry chain including PV materials, PV modules, PV components, PV systems and inverters. Provide third-party technical services throughout the life cycle of PV power stations, such as PV station address and light resource evaluation, design evaluation and optimization of power station quality and performance testing, PV system certification, evaluation and certification of power station operation and maintenance capabilities. |
| 12 | China General Certification Center (CGC) | China | IEC 61215\IEC 61730\GB 37408\GB 37409\NB 32004\CGC-GF063, etc. | Business includes design optimization of PV power stations, optimization of key equipment selection, certification of key equipment such as PV modules, inverters, batteries, etc., acceptance testing of PV power stations, and post-evaluation of PV power station performance. |
| 13 | Shanghai Ingeer Certification Assessment Services (ICAS) | China | IEC 61730\IEC61215\GB 50794\CQC 9102\CNCA 0004, etc. | PV power plant inspection and acceptance, PV module certification and laboratory testing services, key equipment manufacturing supervision services, PV power plant due diligence and evaluation services. |
| 14 | Power (Beijing) Certification Centre (PCCC) | China | IEC 60904\IEC 61215\IEC 61730\GB 34396\CPIA 0006 etc. | Certification business for PV power generation and transmission systems and accessory products such as PV cells, PV cell modules, PV junction boxes and PV connectors, PV combiner boxes, back sheet, circuit breakers, etc. |
| 15 | Credit Builder Certification Wuxi (CBC) | China | IEC 62109\NB 32004\CNCA 0006 etc. | Certification of PV modules, independent PV systems, back sheet, battery performance, grid-connected inverters |

### 3.3.2. IEC Mutual Recognition System

The IEC Conformity Assessment Board (CAB) was established on 14 February 1997. It is composed of four major certification systems by recommending personnel from 12 IEC member states. Its main responsibility is to formulate conformity assessment policies from the perspective of conducive to the development of international trade. Among them, IECEE (IEC System for Conformity Assessment Schemes for Electrotechnical Equipment and Components) and IECRE (IEC System for Certification to Standards Relating to Equipment for Use in Renewable Energy Application) are related to photovoltaic conformity assessment. The predecessor of IECEE was the European Commission for Qualified Inspection of Electrical Equipment (CEE), which was established in 1926. Later, it merged with IEC into IECEE, and promoted the regional mutual recognition system already implemented in Europe to the world. Its business includes 23 categories of electrical and electronic equipment safety, quality, efficiency and overall performance and testing services, of which the 19th is photovoltaic products and components. IECEE-CB is a mutual recognition system between IECEE members, which can realize a one-stop service of one test, one certificate, and international mutual recognition. China is an important member of the IECEE-CB system. Up to now, a total of 39 laboratories in China have passed international peer review to become IECEE-CB testing laboratories, such as CQC Nanjing Testing Center and China Electric Power Academy.

IECRE was established in 2014 and has a solar energy working group under it. Its main tasks are to develop high-quality international standards, establish and operate a

globally unified renewable energy certification system, promote widespread mutual trust in certification results on a global scale, and promote the facilitation of international trade. It effectively eliminates the national differences and standard differences of photovoltaic systems, and breaks through the barriers of the quality inspection of IECRE member countries in power plant investment and financing transactions. It also helps the insurance companies to qualitatively evaluate the quality of photovoltaic systems and quantitatively evaluate the power generation of the system. The certification testing requirements and testing process under the current mutual recognition mechanism of IECRE are shown in Figure 4.

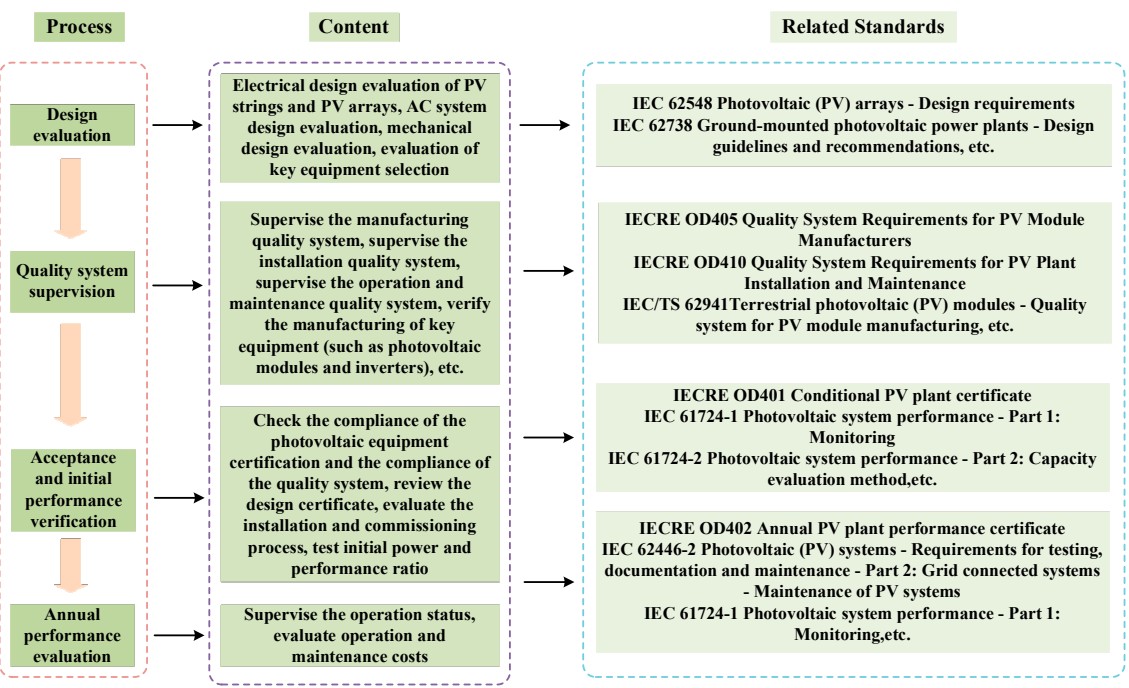

**Figure 4.** IECRE certification process and requirements.

In August 2014, CNCA formally represented China to participate in IECRE and became a founding member. In 2015, China Quality Certification Center (CQC) was accredited by IECRE and obtained the qualification as an IECRE certification body and inspection agency. In 2017, China General Certification Center was also accredited and certified by IECRE, and passed the VDE review to become an IECEE laboratory in 2019. The entry of Chinese institutions signifies that China's certification and testing capabilities have reached the internationally recognized level.

*3.4. Standard Construction of Photovoltaic Industry*

IEC Solar Photovoltaic Energy Systems Standardization Technical Committee (IEC/TC82) is an international technical organization specializing in the formulation and revision of international standards in the photovoltaic field. The secretariat is located in the United States, the current chairman is Michio Kondo of the Japan Institute of Industrial Technology, and the vice chairman is Zheng-Xin Liu, a researcher at the Shanghai Institute of Microsystem and Information Technology, Chinese Academy of Sciences.

At present, IEC/TC82 has seven working groups, the organizational structure is shown in Figure 5, and the business scope of each working group is shown in Table 2.

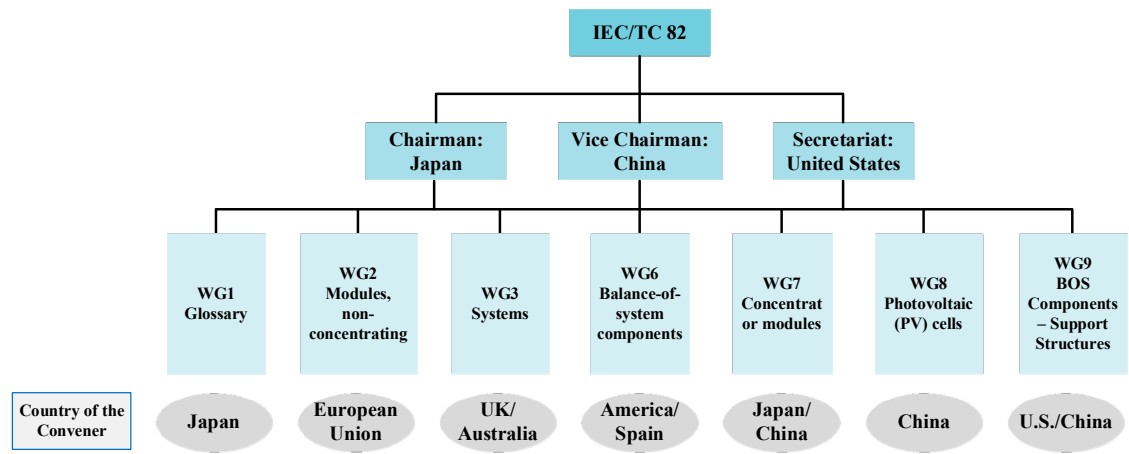

**Figure 5.** IEC/TC82 organization.

**Table 2.** IEC/TC82 Working Groups and Tasks.

| Working Group | Title | Task |
|---|---|---|
| WG1 | Glossary | To prepare a glossary. |
| WG2 | Modules, non-concentrating | To develop international standards for non-concentrating, terrestrial photovoltaic modules. These standards will be in the general areas of photoelectric performance, environmental test, quality assurance and quality assessment criteria. |
| WG3 | Systems | To give general instructions for the photovoltaic system design, construction and maintenance. |
| WG6 | Balance-of-system components | To develop international standards for balance-of-system components for PV systems. These standards will be in the general areas of performance, safety, environmental durability (reliability), quality assurance and quality assessment criteria. |
| WG7 | Concentrator modules | To develop international standards for photovoltaic concentrators and receivers. These standards will be in the general areas of safety, photoelectric performance and environmental reliability tests. |
| WG8 | Photovoltaic (PV) cells | To develop international standards for non-concentrating, terrestrial photovoltaic cells. These standards will be in the general areas of photoelectric performance, environmental test, quality assurance and quality assessment criteria |
| WG9 | BOS Components—Support Structures | To develop international standards for photovoltaic support structures. These standards are in the general areas of safety, design qualification, engineering integrity, durability, and verification testing. |

In recent years, China's photovoltaic industry has made full use of its own technological foundation and industrial supporting advantages to develop rapidly, the scale of the industry has expanded rapidly, and the industry standards have been further improved. In 2009, Standardization Technical Committee of China Photovoltaic Industry Association (CPIA) revised the photovoltaic standard classification and standard system structure, and China's photovoltaic standard system has basically been formed. In March 2016, CPIA formally established a Standardization Committee [44] and began the formulation of group standards. Up to now, 32 group standards have been issued. With the participation of various associations, the photovoltaic standard system is becoming more and more complete. In 2017, the Ministry of Industry and Information Technology of China issued the "Technical Goals for Comprehensive Standardization of the Solar Photovoltaic

Industry" [45], which initially formed the framework of the photovoltaic standard system. In August 2021, the China Electronic Technology Standardization Research Institute and Secretariat of CPIA led the organization of the revision of the "Solar Photovoltaic Industry Comprehensive Standardization Technical System". On the basis of the 2017 standard framework, two major directions, "green photovoltaics" and "smart photovoltaics", have been added. Additionally, many small categories such as electrical components and devices, supporting structures and components have been added and adjusted. The adjusted schedule of photovoltaic standards is divided into nine directions and a total of 761 standards, forming the "14th Five-Year Plan" Photovoltaic Industry Comprehensive Standardization Technical System Framework Diagram (Draft), as shown in Figure 6.

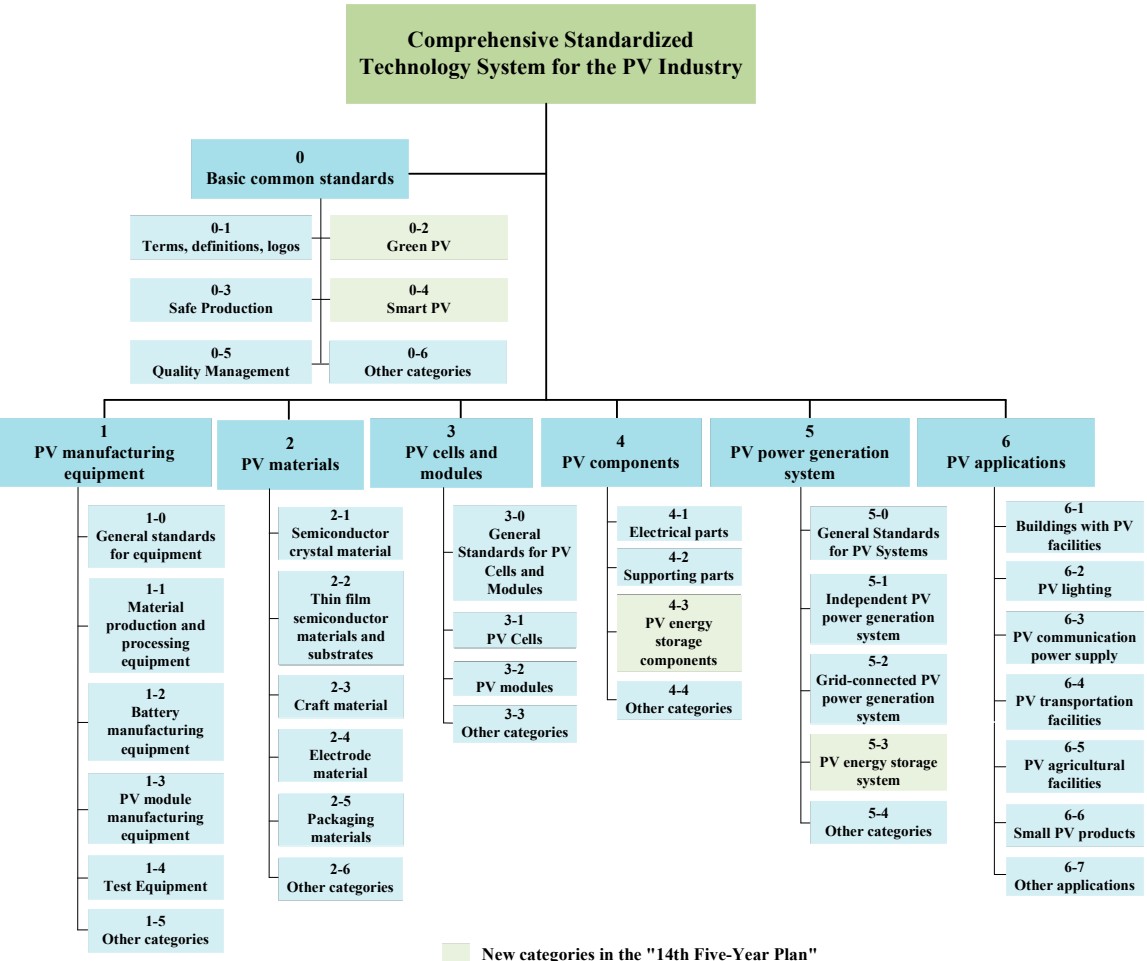

**Figure 6.** "14th Five-Year Plan" Photovoltaic Industry Comprehensive Standardization Technical System Framework Diagram.

In recent years, China has, increasingly, actively participated in the formulation of IEC international standards. In May 2011, the IEC/TC82 annual meeting was held in Shanghai. China submitted three standard proposals to IEC/TC82 for the first time, and was responsible for the review and appraisal of three international standards. In the following few years, China has submitted a number of proposals and a number of standards have been published (see Table 3). Among them, the IEC TS 63217:2021 "Utility-interconnected photovoltaic inverters—Test procedure for over-voltage ride-through measurements" prepared by China Huaneng Group was successfully released and implemented. This standard proposes the high-voltage ride-through test method of photovoltaic inverters at the level of international standards for the first time, which provides countries around the world with a unified and complete test basis for high-voltage ride-through of photovoltaic inverters,

and further improves the stability of photovoltaic grid-connected. The issuance of this standard fills the gap in the field of high-voltage ride-through standards for photovoltaic inverters worldwide.

**Table 3.** Published IEC international standards dominated by China.

| Index | Standard Number | Standard Name | Release Year |
|-------|-----------------|---------------|--------------|
| 1 | IEC 62788-1-6 | Measurement procedures for materials used in photovoltaic modules—Part 1–6: Encapsulants-Test methods for determining the degree of cure in Ethylene-Vinyl Acetate | 2017 |
| 2 | IEC 62805-1 | Method for measuring photovoltaic (PV) glass—Part 1: Measurement of total haze and spectral distribution of haze | 2017 |
| 3 | IEC 62805-2 | Method for measuring photovoltaic (PV) glass—Part 2: Measurement of transmittance and reflectance | 2017 |
| 4 | IEC 63202-1 | Photovoltaic cells—Part 1: Measurement of light-induced degradation of crystalline silicon photovoltaic cells | 2019 |
| 5 | IEC 62910 | Utility-interconnected photovoltaic inverters—Test procedure for under voltage ride-through measurements | 2020 |
| 6 | IEC 61327 | Utility-interconnected photovoltaic inverters—Test procedure for over voltage ride-through measurements | 2021 |

*3.5. Market Surveillance*

The later stage of the development of the photovoltaic industry is the stage of complete marketization. The government no longer provides financial support to the photovoltaic industry, and enterprises and users have become economic entities in the market [46]. It is necessary to handle the relationship between the government and the market through market supervision, and to coordinate it through cost supervision and market management. Germany, Japan, and the United States are already at the stage of marketization. Since 2017, Germany has no longer used the government to set prices for purchases, but instead provides subsidies through market bidding. Since 2021, Japan has implemented a fixed (premium) subsidy policy—Feed-in Premium (FIP), and photovoltaic power generation is connected to the grid through bidding without distinction from other energy power-generation technologies in accordance with the rules of the power market. The United States supports the development of the photovoltaic industry through taxation policies, and the government does not provide subsidies. However, its developed financial market and extensive participation of venture capital institutions have provided sufficient financial support for the photovoltaic industry and promoted the stable development of the photovoltaic market.

However, the development of the photovoltaic industry in many developing countries still relies on government subsidies, or is transitioning from the technology demonstration stage to the early stage of marketization. The Indonesian government plans to invest $15.7 billion in new energy by 2025. India has launched a $600 million subsidy program for its own photovoltaic manufacturing, and will invest about $150 billion to $200 billion in power transmission and transformation infrastructure and energy storage in the next 8.5 years to improve photovoltaic grid connection. After China has drastically reduced the scale of domestic subsidies, market growth will return to rationality. Photovoltaic electricity prices have fallen by more than 67.8% in ten years, and finally achieved grid-connectedness at par, ensuring fair competition in the photovoltaic market.

The levelized cost of energy (LCOE) is usually used to measure the unit power generation cost of the entire project cycle of photovoltaic power generation. LCOE is determined by factors such as initial investment, operating costs, taxes, and power generation.

According to the latest IRENA report, between 2010 and 2019, the global LCOE of utility-scale photovoltaic power plants dropped sharply by 82%, from $0.378/kWh in 2010 to $0.068/kWh in 2019. All major markets, including China, India, Japan, the US,

Europe, and South Korea, saw significant reductions in installed costs. The economics of photovoltaic power generation have increased significantly. Photovoltaic module prices fell by more than 90%, which is the direct cause of the decline in the LCOE of solar power generation. Given the decline in the cost of each link in the photovoltaic industry chain, coupled with the continuous improvement of module efficiency, by 2030, the global photovoltaic LCOE will still show a downward trend year by year.

Due to the small scale of rooftop photovoltaic projects, the total installed cost is generally higher than that of utility-scale photovoltaic power station projects. However, between 2010 and 2019, the cost of rooftop PV projects also fell by 47–86%. Cost reduction and policy support have made photovoltaics enter the household era, but there are still many problems in the construction of distributed photovoltaics, especially rooftop photovoltaics. Taking China as an example, some photovoltaic installation companies failed to perform their duties in accordance with the contract after installing photovoltaics, and did not perform operation and maintenance of photovoltaic power plants. There are also some unscrupulous businesses that promote photovoltaic loans by exaggerating their profits and driving up installation costs, seriously jeopardizing the interests of installers and causing a lot of market chaos. Faced with these problems, it is necessary to strengthen market supervision. It is also necessary to pay attention to the interest disputes that may arise in the process of installation, operation and maintenance. Solutions such as strictly reviewing the qualifications of photovoltaic loans and achieving transparent and checkable photovoltaic installation prices will help to ensure the stable and orderly development of the photovoltaic market.

## 4. Suggestions

### 4.1. At NQI Level

The NQI system has been widely used around the world, which has further improved the quality of products in developed countries such as the United States. Countries such as Spain, Uganda, Thailand, Ukraine, Brazil, and Africa are also actively constructing NQI and improving their own NQI level. In recent years, China has made remarkable achievements in the overall construction of NQI, but the current NQI system performance still lags behind that of developed countries. For China and other countries that are building the NQI system, possible solutions to the problems encountered in the construction are proposed:

1. Pay attention to the combination of theoretical research and practical application. It is necessary to organically integrate the metrology, standards, and conformity assessment in the NQI system with the existing enterprise quality management system. Actively organize and guide enterprises and projects to develop NQI and follow NQI-related requirements [47], making NQI construction more viable and targeted.

2. Pay attention to national strategic needs. Determine the current development direction of the national NQI through top-level design and overall planning. Incorporate the current economic and social development needs, such as the "carbon peaking and carbon neutrality goals", into the scope of NQI research, and form a scientific and reasonable NQI construction strategy, so that the NQI system can maximize its effectiveness.

3. Pay attention to the construction of NQI. The essence of NQI is to serve quality management, serving people's lives, industries, and enterprises. It is necessary to guide and promote the implementation of the NQI, improve the NQI evaluation system, and ensure that the NQI better serves the country, society and consumers.

4. Actively participate in international exchanges and cooperation. The construction of the NQI system has received widespread attention from all countries in the world, and the practical experience and design ideas of NQI in all regions of the world are worth learning and researching. Actively participating in international conferences and participating in the construction of the international NQI system can help the NQI system that is under construction to achieve advancement and integration.

*4.2. Recommendations for the Development of NQI System in Photovoltaic Industry*

Under the current carbon emission targets of various countries, the solar power generation industry should strengthen quality supervision and establish and improve the photovoltaic NQI system. The various elements of NQI are closely linked to each other. If the metrology cannot be traced to the source, neither the standard nor the test has global application value. Without the establishment of a standard system, tests related to conformity assessment cannot be carried out. Products that are not certified cannot enter the domestic and foreign markets. If the market price is not adjusted in time, it will affect the final implementation of the project. All photovoltaic departments should work closely with each other to complement each other, jointly build a scientific and complete photovoltaic NQI system, and realize the integrated development of all elements in the NQI system. In addition, it is necessary to accelerate the construction of an integrated NQI service platform to provide convenient services for the photovoltaic industry. It is also necessary to achieve full coverage of standards, conformity assessment, metrology and market supervision, to ensure the reliable implementation of photovoltaics from production to grid-connected power generation, and to improve the quality of the industry.

There are some development suggestions for each elements of the photovoltaic industry NQI system:

1.  In terms of metrology: (1) Strengthen the metrology service of key parameters and key equipment, formulate standard battery measurement and standard component measurement procedures, and build empirical test bases and public test platforms. (2) Strengthen on-site measurement and test services, especially outdoor photovoltaic measurement, improve the adaptability and flexibility of metrology methods and equipment, and provide customers with convenient services. (3) Provide measurement training services for technical personnel for enterprises. Research metering schemes suitable for distributed photovoltaics and develop more accurate electrical energy measurement instruments.

2.  In terms of standards: (1) Actively organize domestic experts to participate in meetings organized by IEC, and actively speak on standards proposed by member states to enhance international discourse power. (2) Pay attention to the development of international standards, increase communication with the standard-setting group, and summarize and highlight the revision and reporting direction of standards. (3) Realize the integrated and coordinated development of the photovoltaic industry from technology, products, applications to standards and testing, make full use of standards, and improve product quality. (4) It is necessary to speed up the formulation of safety standards and grid-connection requirements related to distributed photovoltaics, especially rooftop photovoltaic projects, starting from the aspects of design, construction, operation and maintenance. Especially in terms of design, it is essential to comprehensively consider site selection, electrical fire protection, engineering construction, etc., and set relevant specifications for the quality of engineering construction. In particular, China must give full play to its leading technological advantages and export volume advantages, transform technology into standards, lead the formulation of international standards, and enhance international competitiveness.

3.  In terms of certification, accreditation, inspection and testing: (1) Equipment should be optimized, testing capabilities should be enhanced, and the international credibility of testing data should be improved. (2) Form a certification and testing system covering the entire photovoltaic industry chain and the entire life cycle of photovoltaic equipment. (3) To increase grid-connected testing and certification capabilities, mandatory testing should be implemented for key content such as system security and actual operating efficiency. (4) For distributed photovoltaics, it is necessary to strengthen mandatory regulations such as leakage protection, arc detection protection, and rapid shutdown of photovoltaic modules.

4.  In terms of market supervision: (1) Strengthen the detection of changes in market costs and prices. In recent years, some silicon wafer manufacturers have been delib-

erately hoarding stocks, causing the price of raw materials to rise. It is necessary to strengthen the monitoring of prices and construction costs and report them to the National Energy Administration in a timely manner as the basis for industry policy adjustments. (2) Strengthen market supervision. Intensify the review of photovoltaic module suppliers and operation and maintenance companies to ensure the quality and warranty period of household photovoltaic products. Severely crack down on arbitrary charges in the construction of rooftop photovoltaic power stations. (3) Set up a standardized grid connection process. Help rooftop photovoltaic installers to connect to the grid easily and smoothly. Realize the transparency of the prices of all parts of the photovoltaic industry to help installers get the most benefits.

## 5. Conclusions

This paper first introduces the NQI in detail, and the focus of the paper is to establish the photovoltaic NQI system. The system can help the photovoltaic industry to build comprehensively, and make overall planning in terms of standards, measurement, and conformity assessment. It can help to avoid unbalanced development that hinders the overall development of the photovoltaic industry, and can also discover the current shortcomings of the photovoltaic industry from the top-level design, and avoid possible problems in time. At present, the photovoltaic NQI system and its various parts have not been fully established. The paper also proposes solutions to some problems in the construction of the NQI system and the NQI construction in the photovoltaic industry.

Although the photovoltaic NQI system has been established in this paper, due to the limited capacity and research time, the specific implementation of the system construction and the coordination methods of various departments cannot be studied in depth. The introduction of the various components of the photovoltaic NQI is also based on the existing information that can be found. The situation in the countries mentioned has not been studied in depth, and the latest unpublished information is, thereby, not included.

**Author Contributions:** Introduction, H.-F.X. and C.-H.N.; background, H.-F.X., Q.-W.C. and Z.-Y.Y.; system architecture, R.S. and H.-F.X.; suggestions, R.S., H.-F.X., C.-H.N., Q.-W.C. and Z.-Y.Y.; writing—review and editing, R.S. and H.-F.X.; supervision, C.-H.N., Q.-W.C. and Z.-Y.Y.; project administration, H.-F.X., C.-H.N., Q.-W.C. and Z.-Y.Y. All authors have read and agreed to the published version of the manuscript.

**Funding:** This research was funded by China Huaneng Group Headquarters Technology Project "Grid-connected photovoltaic system control technology research and grid-connected photovoltaic inverter high voltage ride through test procedures international standard compilation", grant number: HNKJ21-HF263.

**Conflicts of Interest:** The authors declare no conflict of interest.

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
