# Peer review of "National Quality Infrastructure System and Its Application Progress in Photovoltaic Industry"

_electronics, doi:10.3390/electronics11030426_

Round 1

Reviewer 1 Report

The authors described the NQI system as well as the system of the photovoltaic industry, which is a meaningful topic. The manuscript is well designed and the structure flows smoothly. All the arguments are well described and supported. In a word, it is a well prepared work that is worth to be published.

Author Response

Thank you for your appreciation and recognition!

And I have corrected structural and grammatical errors in some sentences.

Reviewer 2 Report

This is a good piece of writing which is novel and of interest. I have some minor suggestions only.

  • Line 68. Literature [9]. Perhaps cite the author's names? 'Moljevic [9]..' Please revise the whole paper to be consistent
  • Line 71 . change to 'issued'. remove hyphen
  • Line 82. 'existing literature lacks a detailed 82 description of the relationship, significance and development status of the various elements in the NQI system'. Can you clarify? And also, perhaps provide a table to demonstrate the previous literature summary? This will greatly help the readers
  • Line 110-114. Perhaps you can revise the paragraph to mention the Sections e.g. 'Section 2 describes xxx, Section 3 covers yyy'.
  • Figures 2-6. Can you enlarge the font in the figures for better visibility?
  • Line 309-311. Voc, Jsc etc, perhaps put in paragraph since these are the variable parameters.
  • I like Table 1. Good job.
  •  

Reviewer 3 Report

National Quality Infrastructure System and its Application Progress in Photovoltaic Industry is presented. There are some major issues that must be addressed.

This paper reasonably well written. However, some of the sentence structuring is clunky and there are some grammatical errors. The writing in this paper can be improved .

The abstract is lacking information on the results of the study.

Even though several papers are referenced and briefly summarised in section 1, it is difficult to get an appreciation for where this study lies in the context of previously published work from the literature review. A review table would provide a concise analysis which would help the reader.

The novel contribution of the paper must be clearly defined. “The framework of NQI is sorted out”, that exactly constitutes sorting out of this framework?  “several development suggestions are put forward” are these the main contribution? I’m finding it difficult to see how this paper will inform readers and significantly add to the field. Much of the content of this paper could be considered matter of opinion rather than matter of fact.

The aims and objectives of the paper should be explicitly stated at the end of section 1. Bullet points would help. It is too ambiguous in its current form.

Most of section 2 is just a general history of national standards. This information is already openly available.

There should be a critical analysis of the proposed NQI system for the photovoltaic industry.

Cognate works should be compared and discussed in the discussion section. How do your suggestions compare with those of previous studies?

The limitations of this work should be stated.

In the conclusion, avoid giving a general summary. Make sure to give actual conclusions.

This paper may be more suited to the format of a review, or survey paper.

Author Response

Thanks for your suggestions, modifications have been made according to the problems you pointed out.

Round 2

Reviewer 3 Report

Adequate changes have been implemented. The authors have followed the recommended instructions. The quality of the paper has been improved.